# Physical Activity, Mental Health and Wellbeing during the First COVID-19 Containment in New Zealand: A Cross-Sectional Study

**DOI:** 10.3390/ijerph182212036

**Published:** 2021-11-16

**Authors:** Wendy J. O’Brien, Claire E. Badenhorst, Nick Draper, Arindam Basu, Catherine A. Elliot, Michael J. Hamlin, John Batten, Danielle Lambrick, James Faulkner

**Affiliations:** 1School of Sport, Exercise and Nutrition, Massey University, Auckland 0632, New Zealand; c.badenhorst@massey.ac.nz; 2School of Health Sciences, University of Canterbury, Christchurch 8140, New Zealand; nick.draper@canterbury.ac.nz (N.D.); arindam.basu@canterbury.ac.nz (A.B.); 3Department of Tourism, Sport, and Society, Lincoln University, Lincoln 7647, New Zealand; catherine.elliot@lincoln.ac.nz (C.A.E.); michael.hamlin@lincoln.ac.nz (M.J.H.); 4School of Sport, Health and Community, University of Winchester, Hampshire SO22 4NR, UK; John.Batten@winchester.ac.uk (J.B.); James.Faulkner@winchester.ac.uk (J.F.); 5School of Health Sciences, University of Southampton, Southampton SO17 1BJ, UK; D.M.Lambrick@soton.ac.uk

**Keywords:** coronavirus, pandemic, exercise, depression, anxiety, wellness, physical distancing, lifestyle behaviour change

## Abstract

Strategies implemented worldwide to contain COVID-19 outbreaks varied in severity across different countries, and established a new normal for work and school life (i.e., from home) for many people, reducing opportunities for physical activity. Positive relationships of physical activity with both mental and physical health are well recognised, and therefore the aim was to ascertain how New Zealand’s lockdown restrictions impacted physical activity, mental health and wellbeing. Participants (*n* = 4007; mean ± SD: age 46.5 ± 14.7 years, 72% female, 80.7% New Zealand European) completed (10–26 April 2020) an online amalgamated survey (Qualtrics): International Physical Activity Questionnaire: Short Form; Depression, Anxiety and Stress Scale-9; World Health Organisation-Five Well-Being Index; Stages of Change Scale. Positive dose–response relationships between physical activity levels and wellbeing scores were demonstrated for estimates that were unadjusted (moderate activity OR 3.79, CI 2.88–4.92; high activity OR 8.04, CI 6.07–10.7) and adjusted (confounding variables: age, gender, socioeconomic status, time sitting and co-morbidities) (moderate activity 1.57, CI 1.11–2.52; high activity 2.85, CI 1.97–4.14). The study results support previous research demonstrating beneficial effects of regular physical activity on mental health and wellbeing. Governments may use these results to promote meeting physical activity guidelines in order to protect mental health and wellbeing during the ongoing COVID-19 restrictions and future pandemics.

## 1. Introduction

Engagement in physical activity is a major determinant of health, and when one’s ability to be physically active is restricted, health is compromised [1]. Exposure of humans to the Novel Coronavirus disease 2019 (COVID-19) forced governments around the world to develop containment strategies in attempts to restrict the spread of the virus. A deleterious consequence of such containment strategies is the potential reduction in physical activity opportunities and increased sedentary activities such as use of computers and televisions or working from home (the latter eliminates active transport or active job environments) [2,3,4,5,6,7]. One immediate health risk, as a consequence of lockdowns worldwide, is a negative effect on mental health and wellbeing, especially in individuals who may be at risk of mental health disorders [8]. If lockdowns are continued for longer periods or result in a sustained decrease in physical activity due to behaviour change, then COVID-19 containment strategies may also have a negative effect on cardiometabolic health [1], with a resultant increase in health economic burdens worldwide.

On 21 March 2020, the New Zealand Government instituted a containment strategy known as the 4-tiered Alert Level System (from Level 1 with life as normal but with border restrictions, through to Level 4 with severe containment), that restricted individuals’ access to many services and activities, including physical activity [9]. The most severe alert level (Level 4) was put in place for all of New Zealand on 25 March 2020. This alert level was subsequently lowered to Level 3 on 27 April, and then progressively reduced to Level 1 on 8 June 2020 [9]. Alert Level 4 reduced the ability of individuals to partake in many kinds of physical activity, removing access to organised sport, community-based exercise, fitness centres and community playgrounds, and limited access to public parks. New Zealand residents were instructed by the Government to self-isolate into “bubbles”, defined as the group of people with whom one resides. Moreover, implementing physical distancing (maintaining a minimum distance of 2 m) from others outside of one’s bubble was stipulated to reduce social contact during Level 4 containment. This degree of restriction was likely to have had detrimental effects on physical activity routines and behaviours [2], and consequently on physical and mental wellbeing [10]. One of the largest impacts on individual physical and mental health may have resulted from the closure of facilities, such as gyms and sporting facilities, playground equipment, cinemas, restaurants, sport spectating venues and places of worship.

The New Zealand Government limited outdoor physical recreation to locations in the local neighbourhood which could be accessed by active transport (i.e., by foot or bicycle) rather than requiring public transport or personal vehicles [9]. This meant that residents could use their homes, backyards, local streets and nearby parks in which to be physically active; however, driving for the sole purpose of exercising, e.g., to the beach for a swim, was not permitted. Limitations were also imposed on higher risk activities like mountain biking and surfing which have greater chance of injury, potentially placing undue strain on emergency response personnel needed for the anticipated rise in COVID-19 patients [9]. Unlike in many European countries, the New Zealand Government placed no limitation on the number of times residents could leave their homes to engage in physical activity, which did allow a degree of freedom for individuals to choose their physical activity time, frequency and duration [11].

A recent containment study, using data from 455,404 mobile phone users worldwide, saw a 27.3% reduction in daily step counts (a proxy for physical activity) after 30 days of confinement [5]. A similar study, which collected data on over 30 million customers worldwide by an electronic fitness company (Fitbit) during March 2020, identified a substantial reduction in daily step counts (ranging from 4 to 38%) compared with the same time the previous year (i.e., 2019) [2]. Other researchers have reported a 32% decrease in the physical activity of American adults during COVID-19 containment restrictions [3], with those individuals who were completing strict self-isolation showing even lower physical activity levels. It is reported that increases in physical activity are not only associated with improvements in physical health but are positively associated with subjective wellbeing [12]. This relationship, however, is bidirectional such that physical activity is considered beneficial in supporting behaviours that promote health and wellbeing, reinforcing regular physical activity participation and subsequently aiding positive subjective wellbeing [12]. Thus, despite the containment strategies implemented, the New Zealand Government still encouraged participation in some form of physical activity (e.g., walking around the block) [9]. The effect of maintaining physical activity for mental health and subjective wellbeing during the COVID-19 lockdown has been explored in number of investigations [3,8,10,13]. Research has found that individuals who did not reach physical activity guidelines and engaged in more screen time during the COVID-19 containment restrictions had higher scores of depression and stress than those who exercised more during this period [3]. Pears et al. [13] found that key sociodemographic and health outcomes, as well as sitting time, explained 42% and 27% of the variance in depression and subjective wellbeing scores, respectively. Subgroup analysis has identified inter-individual differences in mental health during containment analysis [8], with some groups demonstrating an improvement in mental health and wellbeing due to the reduction in mundane stress-inducing factors, commuting and workload. However, others (e.g., older adults, those suffering from mental health disorders/low mental health scores, socially deprived, financially stressed) are likely to experience a continued and progressive decline in mental health and wellbeing scores [8]. In these individuals, the impact of reduced physical activity may be exacerbated by declines in mental health and wellbeing, and subsequently lower the intention to exercise, intensifying the deleterious effects on both physical health and mental wellbeing.

At the time of writing (16 months after initial lockdown), New Zealand’s containment strategy has been relatively successful at containing the COVID-19 outbreak, moving from lockdown Level 4 to Level 1 within 11 weeks, and remaining largely free from community transmission since this time. However, the initial response did come with physical activity restrictions, isolation from family and friends and disruption to normal routines, all of which can contribute to poor physical (obesity, cardiovascular disease, bone density loss, lower aerobic capacity) [14] and mental (higher levels of anxiety and stress) [15] health outcomes and subjective wellbeing [12]. Evidence for the relationship between physical activity and mental health during containment strategies throughout the COVID-19 crisis is still emerging. Therefore, information on the impact of lockdown strategies from various global regions may help governments improve future lockdown strategies to minimise or mitigate negative effects on physical and mental health. The aim of this study was to examine changes in physical activity, mental health and wellbeing brought about through the COVID-19 Level 4 lockdown restrictions in New Zealand as compared with pre-lockdown figures.

## 2. Materials and Methods

Cross-sectional data related to the Level 4 lockdown (25 March to 26 April 2020) of government-led containment strategies in New Zealand were collected using Qualtrics online survey software (Qualtrics, Provo, UT, USA). The research was deemed a low-risk notification by Massey University Human Ethics Committee (Approval number 4000022445). Research was conducted in accordance with the Declaration of Helsinki. This study adhered to current epidemiological guidelines (Strengthening the Reporting of Observational Studies in Epidemiology—STROBE) [16]. All participants provided informed consent at the start of the survey. The sample size was unlimited, meaning anyone meeting the eligibility criteria was eligible to participate.

Convenience and snowball sampling (mass emailing, social media and national radio) were employed during the early period (10–26 April 2020) of COVID-19 government mandated restrictions. All adults aged 18 years and older and living in New Zealand during the Level 4 lockdown with access to the online survey were eligible to participate.

The survey took approximately 15 min to complete and collected information on physical activity (International Physical Activity Questionnaire: Short Form [IPAQ-SF]) [17], mental health (Depression, Anxiety and Stress Scale-9 [DASS-9]) [18,19], subjective wellbeing (World Health Organisation-Five Well-Being Index [WHO-5]) [20], and exercise behaviour change (Stages of Change Scale) [19]. Additionally, demographics were collected, including age, gender, living situation, perceived income security, work status (essential or non-essential worker), and whether comorbidities were present and affected physical activity. All items were assessed during the initial Level 4 lockdown, with some items (e.g., stages of change items, meeting physical activity guidelines) also assessed retrospectively to query how attitudes and physical activity levels may have changed from pre- to during lockdown.

The IPAQ-SF is a valid [pooled ρ for comparisons between long and short forms was 0.67 (95% CI 0.64–0.70)] and reliable (ρ = 0.77–1.00) tool [17] developed to measure physical activity. The 7-item short form records the activity “over the last 7 days” with four intensity levels: vigorous intensity, moderate intensity, walking and sitting [17]. Using the IPAQ-SF on large populations has been validated as an acceptable physical activity measurement tool [21].

The DASS is a commonly used self-report scale that assesses symptoms of depression, anxiety and stress [18]. The 9-item DASS-9 questionnaire (empirically-derived version based on the DASS-21 [22]) consists of three subscales (depression, anxiety and stress). The DASS-9 has been shown to have acceptable to excellent concurrent internal consistency [23], 0.72 for the total scale and 0.52, 0.57, and 0.55 for the depression, anxiety, and stress subscales, respectively, while good construct and convergent validity have been reported [24]. Each item was scored on a 4-point severity/frequency scale from 0 (never) to 3 (almost always) to rate participants’ experiences over the past week. The three subscales of the DASS-9 were each cumulatively scored between 0 and 9, with higher scores demonstrating poorer mental health.

The WHO-5 is a short 5-question global rating scale that indicates subjective wellbeing, and has shown good contrast validity [20]. The WHO-5 includes the following items: (i) I have felt cheerful and in good spirits; (ii) I have felt calm and relaxed; (iii) I have felt active and vigorous; (iv) I woke up feeling fresh and rested; and (v) My daily life has been filled with things that interest me. Each of the five items was scored from 0 to 5. The total raw score was translated into a percentage (raw score multiplied by 5) ranging from 0 (absence of wellbeing) to 100 (maximal wellbeing).

Participants were asked to self-report their exercise intentions for two time periods: pre-Level 4 lockdown (February 2020) and during Level 4 restrictions. The following response options were rated according to the Stages of Change Scale [19]: (i) I currently do not exercise and do not intend to start in the next 6 months; (ii) I currently do not exercise but I am thinking about starting in the next 6 months; (iii) I currently exercise a little but not regularly; (iv) I currently exercise regularly but have begun doing so in the last 6 months; or (v) I currently exercise regularly and have done so for more than 6 months. Borrowed from the Transtheoretical Model of Behaviour Change, these statements align with the pre-contemplation, contemplation, preparation, action, and maintenance stages, respectively [25].

The primary outcome measure was self-reported physical activity level, and the independent variables were mental health (depression, anxiety and stress), subjective wellbeing, and exercise intention (pre- and during Level 4 lockdown). The potential confounding variables were demographics, including age, gender, living arrangements, income, and employment (“essential worker” or not). The overarching research question was, “What is the impact of physical activity on mental health and wellbeing during a stringent period of lockdown in New Zealand?”

### Data Analysis

Data gained from the IPAQ-SF were coded and analysed using the recommended guidelines found on the IPAQ website (www.ipaq.ki.se, accessed 13 July 2020). Using the IPAQ scoring system, the total number of days and minutes of physical activity were calculated for each participant in the areas of moderate- and vigorous-intensity activity along with walking and sitting. In addition, total time spent walking and in moderate- and vigorous-intensity activity were converted to continuous variables (MET·min·week^−1^) according to the recommended guidelines and then summed to give total physical activity (MET·min·week^−1^).

The survey data were entered into a Jupyter notebook and statistical analysis was completed on R (Version 3.5.1). Only individuals who completed all survey items were included in the statistical analysis. Surveys with missing data (*n* = 678) were omitted from the dataset. For the IPAQ-SF, the total physical activity data were not normally distributed so were converted into three equal tertiles. All participants were ranked, with the lowest 33% being in the low level, middle 33% in the moderate level and top 33% being in the highest level of total physical activity. Similarly, the total time participants spent sitting (min·week^−1^) was converted into three equal tertiles based on the lowest, middle and highest level of total sitting time, as sitting time was also not normally distributed. Scores for total physical activity and sitting time were then each entered into separate multiple regression models with the lowest levels being compared separately to the moderate level and then the highest level. The DASS-9 was analysed in the regression models using the total score that ranges from 0 to 27 (sum of depression, anxiety and stress scores); where higher scores related to higher overall depression, anxiety and stress scores. The WHO-5 scores were also not normally distributed so a cut-off point of 50 was used to convert the WHO-5 into a binary variable, whereby ≤50 was classified as a lower wellbeing and >50 as a higher wellbeing.

The binarised WHO-5 scores were used as an outcome variable and assessed the impact of tertiles of physical activity as explanatory variables in a series of multivariable logistic regression models. In these logistic regression models, we reported the odds ratios, 95% confidence intervals (CI) and associated *p*-values. However, as both wellbeing and extent of physical activity were likely to be independently impacted by demographic variables (age and gender), comorbid conditions that would limit a person’s physical activity levels, sedentary lifestyle (the time spent sitting as opposed to spent in active movements), exercise intention, and perceived income levels, these were treated as confounding variables and were controlled for in a stepwise series of incremental models. If the magnitude or direction of the effect estimates were to drop or reverse direction, this suggested a confounding variable.

We also assessed the role of an individual’s employment status during Level 4 lockdown. In New Zealand, an “essential worker” was deemed to be an employee who was able to continue conducting work on-site that was essential to the basic operation of the country, i.e., workers from supermarkets, hospitals, emergency services, police, certain production industries and the like. We assessed the models of the association between physical activities (after adjusting for other covariates) separately for essential and non-essential workers and compared their effect estimates.

## 3. Results

Of the 4007 participants, the mean age was 46.5 ± 14.7 years, with 72.0% female and 80.7% New Zealand European (see Table 1). The majority of participants (63.3%) were between 30 and 59 years old.

Table 1 indicates that living situation both pre- and during lockdown was largely couples (~33%) and two-parent families (~29%). There was, however, a 79% increase in those living with extended family, with 8.2% pre-lockdown increasing to 14.7% during lockdown. This change was likely accounted for mainly by the 10.9% decrease in individuals living alone and a 21.2% decrease in individuals living in flatting or shared household situations.

Before Level 4 lockdown, 78% of participants reported meeting physical activity guidelines, and similarly, 71.4% of participants reported exercising regularly for more than 6 months. The number exercising regularly dropped by 10% to 64.2% during Level 4 lockdown. Those who currently exercised and had begun doing so in the last 6 months shifted from 6.8% before lockdown to 17.4% during lockdown, a 155% increase. Comorbidities affected physical activity engagement for 22% of respondents.

The initial unadjusted binary logistic regression model and the final multivariable logistic regression model are presented in Table 2a,b, respectively, as evidence of the impact of physical activity level (tertile of IPAQ scores) on wellbeing after adjusting for potential confounding variables. The unadjusted estimates suggest that, compared with those individuals who were least physically active during lockdown (i.e., those in the lowest tertiles of IPAQ score), those who had moderate levels of physical activity had higher likelihoods of reporting better mental health status (OR = 3.76, 95% confidence interval: 2.88–4.92). Those who reported the highest levels of physical activity (highest tertile of IPAQ scores), compared with those who had lowest levels of physical activity (lowest tertile of IPAQ scores) were even more likely to report better mental health related to quality of life (OR = 8.04, 95% confidence interval: 6.07–10.7). Hence, physical activity had both a strong effect on wellbeing and the results further suggest that increased levels of physical activity were associated with stronger effects on wellbeing. After controlling for age, gender, socioeconomic status (measured by self-reported sufficiency of income), time spent sitting, comorbidity affecting ability to be physically active and intention to exercise, those who reported moderate levels of physical activity were still more likely to report better wellbeing (middle tertile of IPAQ versus lowest tertile of IPAQ, OR = 1.57, 95% confidence interval: 1.11–2.52). Those who had highest levels of physical activity had even stronger likelihood of having better wellbeing (highest tertile of IPAQ versus lowest tertile of IPAQ, OR = 2.85, 95% confidence interval: 1.97–4.14).

Furthermore, individuals whose comorbid status did not impact their ability to complete physical activities were also more likely to report better wellbeing after adjusting for all other confounders (OR = 2.02, 95% confidence interval: 1.70–2.41). Finally, inference from the analysis suggested that the longer one spent sitting (or the more the tendency of sitting), the less likely they were to report better wellbeing (middle level of sitting compared with least amount of sitting, OR = 0.79, 95% confidence interval: 0.65–0.96). Conversely, those who had the least hours sitting, were more likely to report better mental health (OR = 0.68, 95% confidence interval: 0.56–0.82).

Only 14% of all participants reported being an essential worker (*n* = 567), of whom 64% reported a WHO-5 score over 50. Among non-essential workers (*n* = 2350), 66% had a WHO-5 score over 50, meaning that both essential and non-essential workers had good to excellent overall wellbeing (*p* = 0.347). There were also no statistically significant differences between essential and non-essential workers on DASS-9 stress levels (*p* = 0.697), with 63% and 64% categorised as being stressed during Level 4 lockdown, respectively

## 4. Discussion

Results from our study suggest that, during COVID-19 Level 4 lockdown in New Zealand, there was an apparent dose-dependent relationship between physical activity levels and wellbeing scores; a relationship that remained strong after controlling for age, gender, sitting time, comorbidities, income and exercise intentions. Better wellbeing scores were almost three times more likely among participants reporting the highest amounts of physical activity compared to those with the lowest amount of physical activity. Even participants reporting only moderate levels of physical activity were over one and a half times more likely to report better wellbeing relative to those with the lowest levels. [26]. Although there was a reduction in number of participants exercising regularly during Level 4 lockdown, those who began exercising during the lockdown more than doubled. As with levels of physical activity, there was also increasing likelihood of better wellbeing with greater intention to exercise. These findings align well with those from past studies on physical activity and mental health both before [26] and during the COVID-19 pandemic [3,4,10].

Wellbeing scores improved with age, with the exception of the oldest (80+ years old) age group. Consistent with decades of literature in this area, males tended to report better wellbeing than females and those who did not specify their gender [27,28,29,30]; however, the results were not statistically significant for either. Of note, the 21.8% of participants whose comorbidities impacted their ability to be physically active were twice as likely to report lower wellbeing than those whose comorbidities did not affect their ability to be physically active. Similar findings were reported in research where comorbidity burden in patients with an implantable cardioverter-defibrillator was associated with poor psychological wellbeing and physical health status [31]. Comorbidities may impact on mental health [32] in a way comparable to the Level 4 lockdown restrictions [4], especially if these restrictions resulted in physical activity reductions. Our findings add to the body of evidence that maintaining levels of physical activity at or above national guidelines has benefits for mental health [33,34] with likely positive effects also on physical health [1,35]. More importantly, our findings suggest that enabling and encouraging continued daily physical activity during pandemics and other periods of physical containment is particularly important to support the mental health and wellbeing of individuals and communities.

High sitting time was less likely to be associated with better wellbeing, i.e., those who sat the most during lockdown (once sitting time was split into tertiles) had significantly poorer wellbeing than those who sat the least. Sitting time may have increased for some individuals with the change to working from home and missing out on physical activity associated with commuting to work (cycling, walking to bus stop, walking from parking building) and the lack of distinction between work and home [36]. Previous research indicates similar sedentary and wellbeing trends outside the lockdown setting [37,38,39]. Moreover, those with a greater intention to exercise also reported better wellbeing, although these results were not statistically significant.

The early months of the pandemic brought uncertainty to most people’s lives, with fear of contracting COVID-19, and speculation of widespread job losses and commodity price increases, to name a few. Our results indicated that wellbeing was significantly poorer among participants reporting that they did not have enough money versus those who reported having enough or more than enough money. In this instance, speculated financial implications of the pandemic may have resulted in greater stress and anxiety for those who were already not in a strong financial position to a larger extent than for those with fewer financial worries [40]. Additionally, greater financial security may be correlated with access to fitness equipment (e.g., home swimming pool, bicycles, home-gym equipment) that enabled alternative permissible physical activity options during the lockdown, leading to greater perception of wellbeing.

With regard to living situation, we surmise that changes to physical activity routines may have occurred based on people’s living and surrounding environments. For example, perhaps young adults (e.g., university students) returned to their family home because lectures had gone online and university campuses were closed (pre- versus during lockdown decline in flatting and shared households of 21.2%). Furthermore, older adults may have moved in with family members (decline in living alone of 10.9% pre- versus during lockdown) for support such as shopping (which was discouraged for older adults due to higher contagion risk) or caregiving during the lockdown period. Disruptions to normal routines caused by changes in living situation and environment likely affected where and with whom people were able to exercise. These changes may have positively or negatively impacted the type, duration and enjoyment of physical activities.

We hypothesised that being an essential worker would modify the effect estimate of physical activity on mental health. However, after adjusting for covariates, we found no significant differences between essential and non-essential workers on their mental health according to their WHO-5 and DASS-9 scores. New Zealand had relatively few COVID-19 hospitalisations, and only 18 COVID-19-related deaths up until the end of the study period [41], which is in stark contrast to the UK, US and numerous countries worldwide. It is likely that in our study, only a small proportion of essential workers were frontline medical staff in hospitals overwhelmed with COVID-19 patients and related mortality, but instead most were vital workers who maintained basic operations of the country. It is postulated that having fewer essential workers at the coalface of the pandemic in these high-stress frontline settings may have reduced the impact of stress on the essential worker group as a whole in comparison to other countries during this pandemic. A New Zealand and Australian study by Hays [42] found that employment status impacted quality of life and mental health, with the top mental health concerns of employees (*n* = 3139 professionals surveyed) stemming from financial reasons (40%), return to work anxiety (29%), and isolation in remote work (28%). When examining the New Zealanders alone, 29% reported isolation and loneliness when working from home to be the greatest challenge to mental health and wellbeing during the COVID-19 pandemic. Overall, less than half of participants in that study rated their current mental health and wellbeing as positive, a reduction of 21% compared to pre-COVID-19 levels [42]. Our findings somewhat agree, and suggest that the interruption to what was considered to be “normal working life” prior to the COVID-19 pandemic caused a similar mental health burden on both essential and non-essential workers. More research is needed in this area to examine working from home outside of lockdown periods and whether or not there are long-term consequences.

Among non-essential workers (*n* = 2350), there were significant differences in WHO-5 scores for those reporting the lowest IPAQ tertile scores compared to the middle (*p* = 0.012) and highest tertile (*p* ≤ 0.001). However, among essential workers (*n* = 567), there were no such differences in WHO-5 scores between any of the IPAQ tertiles. This could be a result of the difference in sample sizes with the non-essential workers sample being four times larger, hence having more statistical power. Considering that mental health results were not statistically different between the two groups, perhaps there was increased time and flexibility available for non-essential workers to engage in more physical activity thus having a greater impact on the WHO-5 score. We did not assess how working from home may have affected mental wellbeing or physical activity; however, recently published COVID-19 research from the US found that people working from home due to COVID-19 restrictions reported a reduction in both incidental and structured physical activity as well as increased physical and mental health issues [36]. These health issues were associated with less physical exercise, higher junk food intake, having at least one infant at home, being distracted while working from home, decreased communication with co-workers, increased workload and hours and adjusting work hours around others. Addressing these issues may be important for future lockdown scenarios or if working from home becomes a more acceptable mode of employment and could lead to more suitable home-office environments, greater productivity and better mental wellbeing.

There were a few limitations to consider in this study. First, it must be acknowledged that New Zealand is a small island nation (population 5.1 M), with a vast ocean physically separating it from other countries’ borders. As such, New Zealand is arguably better protected at air and sea ports compared to many other countries, e.g., within Europe. This heightened ability to control the borders could have helped protect New Zealanders not only from COVID-19 exposure but also from the stress associated with contracting the virus. Compared to the management plans of other countries, this may have improved the wellbeing of New Zealand residents when compared to other countries with densely populated cities. Since the initial Level 4 restrictions in New Zealand, a total of 26 deaths related to COVID-19 have been recorded (0.53 deaths per 100,000), the lowest ranking in the OECD [43]. Low population density in New Zealand meant that there was, in most locations, plenty of space for physical separation when engaging in outdoor physical activities so that no limitations on frequency and duration were needed. Perhaps high-density cities such as Tokyo, London and Paris would not have been able to enjoy such spatial freedom when it came to physical activity. As a result, our findings may not be generalisable to nations with high population density and that share multiple borders with other countries, as their ability to allow physical activity levels similar to those afforded New Zealanders during level 4 lockdown may not be possible. Our investigation did consider the impact of a number of covariates that influence physical activity levels and subsequently subjective wellbeing and mental health. Covariate data were obtained from the start of the lockdown period; however, we have limited data from prior to the containment strategy being implemented. As such, we are unable to ascertain the impact of stress alleviation (e.g., absence of commuting and/or reduction in workload) and the positive impact on subjective wellbeing, or if the presence of poor subjective wellbeing and mental health scores were further exacerbated with lockdown. While the results of this study highlight the benefits of physical activity for wellbeing and mental health, we should acknowledge inter-individual differences and the impact of prior mental health and wellbeing as influencers on the results which we obtained. The mixed-sex generalisability of our study is limited by the rather high sample size of females (72%). This is, however, similar to other investigations that have assessed sitting time and the effect on subjective wellbeing and mental health in the UK during the COVID-19 lockdown [13]. The sex distribution in the sample was not unexpected as research has suggested that females are more likely to respond to online research surveys than males [44], especially if the recruitment relies on convenient and snowballing methods. To ensure equal distribution of sex in research surveys, recruitment methods may need to be tailored to specific male cohorts and populations, thus enabling the generalisability of results to the wider population. In addition, our sample had an overall higher proportion of participants who were more physically active than the typical levels reported by the general population in New Zealand, and a high (80.7%) proportion were European New Zealanders. This sampling bias may again be the result of recruitment methods employed by the researchers, and in future research we may suggest a more targeted recruitment method to ensure a sample reflective of the population as a whole. A final consideration was that the survey was available for completion online only, which prohibited participation of those without internet access.

## 5. Conclusions

In closing, our findings add further support to the importance of engaging in regular physical activity, as this is associated with maintaining mental health. Our findings suggest that there was a dose-dependent relationship between physical activity and mental health and wellbeing scores. It is important that, during future crises resulting in lockdowns, governments make concerted efforts to develop physical activity-friendly policies to allow people continued freedom to engage in a preferred duration and frequency of activity so long as appropriate physical distancing and other necessary safety precautions are maintained. Consideration should be given to individuals with comorbidities, poor subjective wellbeing prior to lockdowns, those experiencing financial strain and increased sitting time due to the working from home environment, as all were found to be negatively associated with physical activity and mental health. Providing support for these subgroups in the population may aid in providing a buffer to the negative impacts of physical inactivity on mental wellbeing.

## Figures and Tables

**Table 1 ijerph-18-12036-t001:** Frequencies and percentages of all variables (*n* = 4007).

Variable		Pre-Level 4 Lockdown		During Level 4 Lockdown	
		*n*	%	*n*	%
Age (years)	<29			619	15.45
	30–39			775	19.34
	40–49			910	22.71
	50–59			853	21.29
	60–69			578	14.43
	70–79			250	6.24
	80+			22	0.55
Gender	Male			1087	27.13
	Female			2886	72.02
	Not specified			34	0.85
Essential worker	No			2350	58.65
	Yes			567	14.15
	Not specified			1090	27.20
Comorbidity affecting engagement in physical activity	Yes			873	21.79
	No			2978	74.32
	N/A			156	3.89
Living situation	Alone	515	12.9	459	11.5
	Couple	1379	34.4	1287	32.1
	Two-parent family	1162	29.0	1164	29.1
	Single-parent family	121	3.0	114	2.9
	Extended	328	8.2	588	14.7
	Flatting	501	12.5	395	9.9
	Residential care	1	<0.1	0	0.0
Met physical activity guidelines	Yes	3133	78.1		
	No	874	21.9		
Exercise behaviour	Did not exercise, no intent in next 6 months	42	1.1	31	0.8
	Did not exercise, thinking to start in next 6 months	95	2.4	131	3.3
	Exercising a little, but irregularly	735	18.3	572	14.3
	Exercise regularly, only began in last 6 months	274	6.8	699	17.4
	Exercise regularly, have been for >6 months	2861	71.4	2574	64.2

**Table 2 ijerph-18-12036-t002:** (**a**). Single variable logistic regression model where binarised WHO-5 score regressed on tertiles of IPAQ score (crude odds ratio). (**b**). Multivariate logistic regression model of binarised WHO-5 score on IPAQ scores after adjusting for age, gender, comorbid conditions affecting ability to be physically active, sedentary behaviour (time spent sitting), intention to exercise, and perceived income level.

**(a)**				
**Variable**	**Odds Ratio**	**Lower Limit**	**Upper Limit**	** *p* ** **-Value**
IPAQ Score (Lowest tertile is reference category)				
Middle tertile	3.76	2.88	4.92	<0.001 *
Highest tertile	8.04	6.07	10.7	<0.001 *
**(b)**				
**Variable**	**Odds Ratio**	**Lower Limit**	**Upper Limit**	** *p* ** **-Value**
IPAQ Score (Lowest tertile is reference category)				
Middle tertile	1.57	1.11	2.22	0.011 *
Highest tertile	2.85	1.97	4.14	<0.001 *
Age in years (<29 is reference category)				
30–39	0.89	0.70	1.13	0.345
40–49	0.97	0.77	1.23	0.804
50–59	1.73	1.35	2.22	<0.001 *
60–69	2.63	1.96	3.52	<0.001 *
70–79	4.09	2.61	6.42	<0.001 *
80+	2.19	0.74	6.50	0.159
Gender (Male reference category)				
Female	0.87	0.73	1.04	0.136
Undeclared	0.93	0.40	2.14	0.861
Comorbidity affects PA				
PA is affected vs. not affected	2.02	1.70	2.41	<0.001 *
Time spent sitting (Lowest tertile reference category)				
Middle tertile	0.79	0.65	0.96	0.017 *
Highest tertile	0.68	0.56	0.82	<0.001 *
Exercise intention (Did not exercise, no intent in next 6 months reference category)				
Did not exercise, thinking to start in next 6 months	0.61	0.23	1.63	0.324
Exercising a little, but irregularly	0.89	0.36	2.21	0.798
Exercise regularly, only began in last 6 months	2.07	0.83	5.20	0.120
Exercise regularly, have been for >6 months	2.16	0.87	5.39	0.097
Enough income to meet needs (Not enough reference category)				
Only just enough money	1.47	0.93	2.32	0.095
Enough	2.02	1.33	3.08	0.001 *
More than enough	2.37	1.56	3.62	<0.001 *
Do not know	0.74	0.23	2.37	0.618

Abbreviations: IPAQ, International Physical Activity Questionnaire; PA, physical activity. * indicates statistical significance.

## Data Availability

The datasets used and/or analysed during the current study are available from the corresponding author on reasonable request.

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
