# Peer review of "Physical Activity, Mental Health and Wellbeing during the First COVID-19 Containment in New Zealand: A Cross-Sectional Study"

_ijerph, 2021, doi:10.3390/ijerph182212036_

Round 1

Reviewer 1 Report

The manuscript presents an analysis of physical activity, mental health, and wellbeing during the first stage of home confinement in New Zealand.

I find the manuscript interesting, readable and rigorous; the limitations of the work are concisely pointed out.

The literature points to the effects of house confinement on people's health, and this article contributes to this field by highlighting the importance of physical activity and its benefits on mental health and wellbeing. However, I believe that certain modifications are necessary to improve the final quality of the manuscript.

Introduction

  • Missing references in the following lines: 39-42, 67-78.
  • Line 94-95: They point out that the effect of maintaining physical activity for mental health and subjective wellbeing during the COVID-19 lockdown has been explored in number of investigations, but no corroborating reference appears.

Methodology.

The tools, data collection process and analyses carried out are explained clearly and in detail.

  • Line 199 is in bold, correct.

Results.

The results are detailed in a clear and concise manner.

Discussion

  • The discussion is relevant, but I think it is necessary to include some further reference to the effects of home confinement on mental health, well-being, or physical activity.

Reviewer 2 Report

Very nice discussion!

I'd only suggest reconsidering the eligibility criteria: instead "living in New Zealand at the time of the survey" _ "living in New Zealand at the time of the Level 4 lockdown".

Author Response

The authors agree that the inclusion criteria could have been worded better. The statement mentioned has been amended to read "living in New Zealand during the Level 4 lockdown". 

Reviewer 3 Report

The study evaluated how New Zealand’s COVID-19 lockdown restrictions affected physical activity levels and the relationship to mental health and wellbeing. Study participants (n=4007) completed a survey which included  the International Physical Activity Questionnaire: Short Form, Depression, Anxiety and Stress Scale-9, the World Health Organization-Five Well-being Index, and the Stages of Change Scale. The authors report a positive relationship between physical activity levels, mental health, and wellbeing.  

The manuscript is well written, and the methodology is well conducted. The discussion is interesting and raises appropriate issues, albeit it is slightly lengthy. I have nothing to add as comments and the paper can be published as is.

Author Response

The authors thank the review for their time and consideration of the manuscript and for its acceptance without emendations.

This manuscript is a resubmission of an earlier submission. The following is a list of the peer review reports and author responses from that submission.

Round 1

Reviewer 1 Report

This paper reported that very brief and clear results ("physical activity had both a strong effect on wellbeing and the results further suggest that increased levels of physical activity were associated with stronger effects on wellbeing).

It is required for considering the following points.

First, it is necessary to review the key theoretical variables that can affect subjective well-being. In this study, exercise is emphasized. It is necessary to theoretically review variables such as social relationship, health, and relative deprivation that were mentioned as variables affecting subjective well-being in previous studies.

Second, it is necessary to explain why the independent variables are set as they are now in the analysis model. Why is exercise intention an independent variable?

Third, it is necessary to review the representativeness of the sample. In terms of gender, the number of female respondents is very high.

Fourth, it is necessary to check the common method bias.

Reviewer 2 Report

Dear authors,
thank you for your work. It is a topic of current importance that allows us to reinforce the need for physical activity in improving health. Your paper is quite complete, although I would like to make some suggestions below that could improve the quality of the paper:
Abstract: the abstract is complete and complies with the journal's guidelines. It clearly states the important aspects of the paper.
Introduction: This section could be improved, as although this is a topic in growing development, there are already many works published that focus on the subject of this study, so it would be positive for the quality of this work, to improve this section by including other background information.
Materials and Method: This section does not include the study sample and the research design. The rest of the sections, such as variables, instruments and protocol followed are very well covered and the instruments are also reliable and have already been validated beforehand.
Results: If possible, improve the design of the tables so that they are clearer and easier to understand visually.
Discussion: I consider this section to be complete.
Conclusions: This section should be improved. Draw conclusions according to the objective of the study. The results obtained allow you to extend the conclusions of your study.
Kind regards

Round 2

Reviewer 1 Report

Important variables affecting well-being are still missing. As this is a fundamental limitation of this paper, there is no reason to change the decision in the first round.